# Glucocorticoids of European Bison in Relation to Their Status: Age, Dominance, Social Centrality and Leadership

**DOI:** 10.3390/ani12070849

**Published:** 2022-03-28

**Authors:** Amandine Ramos, Jean-Patrice Robin, Lola Manizan, Cyril Audroin, Esther Rodriguez, Yvonne J. M. Kemp, Cédric Sueur

**Affiliations:** 1Université de Strasbourg, CNRS, IPHC UMR 7178, 67000 Strasbourg, France; amandine.ramos@iphc.cnrs.fr (A.R.); jean-patrice.robin@iphc.cnrs.fr (J.-P.R.); lola.manizan@live.fr (L.M.); cyril.audroin@etu.unistra.fr (C.A.); 2PWN Waterleidingbedrijf Noord-Holland, Postbus 2113, 1990 AC Velserbroek, The Netherlands; esther.rodriguez.gonzalez@pwn.nl (E.R.); yvonne.kemp@ark.eu (Y.J.M.K.); 3ARK Nature, Molenveldlaan 43, 6523 RJ Nijmegen, The Netherlands; 4Institut Universitaire de France, 75231 Paris, France

**Keywords:** *Bison bonasus*, stress, sociality, collective decisions, ruminants, conservation, glucocorticoids

## Abstract

**Simple Summary:**

This study quantified glucocorticoids in faeces of wild European bison and correlated it to different aspects of social status (leadership, age as a proxy of experience, dominance, social centrality) in order to better understand social mechanisms in this endangered species. Measurement of faecal glucocorticoid metabolites could thus be a valuable tool to follow and improve the adaptation and the welfare of herds in the semi-wild and in captivity by long-term following states of animals and changing conditions, according to them.

**Abstract:**

Stress is the body’s response to cope with the environment and generally better survive unless too much chronic stress persists. While some studies suggest that it would be more stressful to be the dominant individual of the group, others support the opposite hypothesis. Several variables can actually affect this relationship, or even cancel it. This study therefore aims to make the link between social status and the basal level of stress of 14 wild European bison (*Bison bonasus*, L. 1758) living together. We collected faeces and measured the faecal glucocorticoid metabolites (FGM). We showed that FGM is linked to different variables of social status of European bison, specifically age, dominance rank, eigenvector centrality but also to interactions between the variables. Preferential leaders in bison, i.e., the older and more dominant individuals which are more central ones, are less stressed compared to other group members. Measurement of such variables could thus be a valuable tool to follow and improve the conservation of species by collecting data on FGM and other social variables and adapt group composition or environmental conditions (e.g., supplement in food) according to the FGM concentration of herd individuals.

## 1. Introduction

Stress is the body’s response to cope with the environment and better survive [1,2,3], except in the cases where stress is chronic. For many social species, an individual’s baseline stress level is generally associated with their dominance status [4]. However, while some studies suggest that it would be more stressful to be the dominant of the group [5,6,7], others support the opposite hypothesis [8,9]. Several variables can actually affect this relationship, or even cancel it, such as the social system of the species, the seasonality of reproduction, domestication, personality, age, or even the context of the study, i.e., in captivity or in the wild [4,10,11,12].

The basic level of stress can also be intrinsically linked to the propensity to be a leader during group movements, i.e., to initiate the departure and be followed [13]. In equines, for example, the stallion usually moves behind the progression; this position allows it to maintain social cohesion by bringing its females together, i.e., herding behaviour [14,15]. A study in Przewalski’s horse species, *Equus ferus przewalskii*, showed a link between the change in the structure of the enclosure, the secretion of glucocorticoids in the male and the lead of collective movements [16] indicating that environmental changes may influence leadership through glucocorticoids concentration. Thus, FGM might be a mediator to cope with external pressures. A growing number of studies attempt to explain the phenomenon of leadership by focusing on individual characteristics (e.g., age, sex, physiological status, temperament, etc.) and on the social relationships that individuals maintain with each other (e.g., dominance, social centrality) (see [17,18,19] for review). However, the majority only focus on the interaction between two or three factors (often age and dominance or dominance and sex) while these factors are often entangled and dependent on internal processes such as glucocorticoids concentration. For instance, in Highland cattle [20,21,22,23] or in other species as elephants, age influenced dominance and with age comes experience of the environment. Moreover, some models in primates succeed in determining dominance owing to age and sex [24] or leadership due to dominance and body size [25]. In Alpine Ibex, dominance, body size and age are correlated [26].

Glucocorticoid concentrations are also linked to social capital but this correlation—positive or negative—depends on the stability of the hierarchy and other parameters as the reproduction state [27,28,29,30]. Mainly, high-ranking individuals have lower concentrations of glucocorticoids when the hierarchy is stable but higher concentrations when it is unstable. Having a certain number of relationships is good but too many may lead to negative effects on health. In Plains Bison (*Bison*, L. 1758), individuals have linear hierarchies and males tend to be more aggressive when growing older, using physical aggression. High-ranking males defend their position leading to stress of domination (e.g., higher ranks are more stressed). However, females rely more on non-physical aggression, meaning intimidation and this leads to stress of dominance (lower ranks are more stressed) [31]. As for other ungulates, rank is often linked to age, as to body weight, also linked to age [32]. Many studies stipulated that stress and glucocorticoids concentration are interchangeable whilst this is not always the case [33,34,35]. Moreover, while an acute increase in glucocorticoids in response to a stressor might indicate that an individual is dealing with more stressors, long-term glucocorticoid profiles are not simply higher in more ‘stressed’ animals, and lower in less ‘stressed’ animals [36]. This is why in the rest of the paper, we only mention faecal glucocorticoid metabolites (FGM) concentration.

This study therefore aims to make the link between leadership, age, dominance, social centrality and the FGM concentration of individuals. We focused our study on semi free-roaming European bison (*Bison bonasus*, L. 1758), a wild ungulate living in groups with an average of 20 individuals [37,38]. The only study reporting the interaction between glucocorticoid concentrations, individual characteristics and social structure in European bison was performed on captive herds and showed that glucocorticoid concentrations were not influenced by sex or age [39]. On the basis of behavioural observations made on leadership and sociality in this species [40,41,42] and the literature on the subject, we made the following hypothesis: the oldest females in the herd, which are the preferential leaders and the more dominant and central individuals, should have a lower average rate of FGM compared to the other group members. Indeed, in stable hierarchy or stable group composition, individuals with higher dominance rank and higher social centrality often show lower FGM concentration [17,18,19,20]. Dominance is also linked to spatial position of group members and low or middle rank individuals are often at the group spatial periphery, which may increase their FGM concentration [31]. Moreover, leadership is associated with a more independent behaviour which is linked to lower concentrations of FGM [43,44,45,46].

## 2. Materials and Methods

### 2.1. Study Area

Our study was conducted in the Kraansvlak area, a coastal dune area that is part of the Zuid-Kennemerland national park (3800 ha), the Netherlands (52°23′16.1″ N, 4°34′41.3″ E). This area was 330 ha and consisted of six main habitats: sand dunes, grasslands, deciduous forest, pine forest, shrubbery areas and scattered water points. No natural predator of bison, i.e., grey wolf (*Canis lupus*, L. 1758) and brown bear (*Ursus arctos*, L. 1758), was present in the Kraansvlak area.

Kraansvlak is a nature core area within the National Parks where access to visitors is limited to guided excursions year round and a hiking trail accessible for the public during the months from September to March. The rest of the year, the free-access walking trail remains closed. Only researchers and park rangers track the animals to monitor or study them.

### 2.2. Study Subjects

During the study period, the herd of European bison consisted of 14 individuals. Bison were identified at an individual level (Table 1). The herd included eight adult females aged 4 to 15 years, two subadult females of 2 and 3 years old, two juvenile males both 10 months old and two juvenile females of 8 and 9 months old. As for many ungulates, bison show sexual segregation with males only coming in the herd during the reproduction season [47,48,49]. Even in semi-wild herds having a male, this one is often isolated. The kin relationships between all individuals were known (Figure 1). Unlike numerous European bison populations living in natural reserves, the herd of Kraansvlak is not supplemented with food. Therefore, this herd shows the most natural foraging behaviour throughout the different habitats all year round. More details are given in Ramos et al. [42] and Ramos et al. [38].

### 2.3. Data Collection

Throughout the data acquisition period (from 5 April 2016 to 22 June 2016), we observed the bison herd from 9:30 to 16:30, for an average of 4 h per day depending on the climatic conditions as well as the time required to locate the group in the area. The total observation time for the entire study period was 156 h. We observed the bison at a minimum distance of 50 m and a maximum distance of 60 m. To avoid stressing the animals and to ensure that our presence did not interfere with natural activity patterns, this distance sometimes decreased slightly due to terrain geography or herd movement direction.

We recorded social relationships of individuals, dominance interactions, movement initiations (leadership) and collected faeces in order to obtain a measure of FGM concentration. Social and dominance data were collected using CyberTracker 3.0 Software (CyberTracker Conservation, Cape Town, South Africa) [50], installed on a Samsung Galaxy Tab3 Lite (Samsung, Seoul, South Korea). Leadership data were recorded with a HDR-CX320E Sony camera recorder (Sony, Tokyo, Japan).

Social relationships: Every 5 min, the identity of the nearest neighbour for each individual was recorded by the instantaneous sampling method [51]. When it was not possible for the observers to distinguish only one nearest neighbour (focal individual placed at the same distance of several individuals), the identities of these different neighbours were recorded. We used the eigenvector centrality as a measure of social centrality. This measures the number and strength of associations of an individual, whilst taking into account the strength of associations of the individuals with whom it is itself associated and is often described as popularity [52].

This index was calculated using the nearest-neighbour matrix implemented in SocProg 2.7 (Hal Whitehead, Dalhousie University, Halifax, NS, Canada) [53]. More details are given in [38,42].

Dominance interactions: All unidirectional agonistic interactions were recorded using the continuous sampling method [51]. For each interaction (see [38]), the identity of the emitter and the receiver was noted. Agonistic interactions were carried over an asymmetrical matrix to calculate dominance scores of individuals. This was analysed using SocProg 2.7 Software to measure the linearity of the hierarchy (h′ = 0.949, *p* < 0.0001, 10,000 permutations; [54]) and to obtain a dominance index for each animal, the Modified David’s Score (MDS). From these MDSs, a dominance score, ranging from 1 (most dominant) to 14 (least dominant), was attributed to each bison. The dominance never changed during the observation period. The linearity of the dominance is very high showing few bidirectional directions and few aggression from subordinate to dominant individuals. Never the subadult or juvenile bison show aggressive behaviours towards the adult ones. Up on 956 interactions, 141 were horn punches, 278 were threatening, 526 were supplanting behaviours and only eleven were avoiding behaviours. Detailed interactions are given in Zenodo repository: https://doi.org/10.5281/zenodo.6348624 (accessed on 14 February 2022).

Leadership: We recorded the initiation phase of spontaneous collective movements with the continuous sampling method [53] by triggering the camera as soon as an individual moved away from the herd. The initiation of a movement was defined as the departure of the first individual (i.e., the initiator) out of the group diameter, for at least 20 steps in a constant direction (±45°), with its head raised [40,42]. This head position has been described to be a good cue of individual motivation to change location [55,56]. An initiation was finally considered to be a success when the initiator was followed by at least one congener [40,42]. If no bison followed the movement within the 15 min time window, the initiation was considered as a failure. The leadership score was then calculated by dividing the number of successful initiations per individual per the total of successful initiations (see [42] for the complete method).

### 2.4. Measures of Faecal Glucocorticoid Metabolites (FGM)

The faecal samples were collected during the observations, meaning from concrete observed individuals, just after defecation. Because bacterial enzymes are reported to increase or decrease concentrations of FGM if samples are not frozen shortly after sampling, only faeces dating from a maximum of 2 h after defecation, were taken [57,58,59]. For this, sterile 5 mL Eppendorf tubes were used, as well as disposable sampling spatulas. At least, one sample per day was taken, and all samples were identified on an individual level. The filled tubes were labelled with the individual ID and date and then immediately stored in a coolbag (containing an ice pack) in the field for a maximum period of 3 h, then they were stored at −20 °C until laboratory analyses took place. At the time of sampling, we noted the environmental conditions (ambient temperature, sunshine or cloudiness, presence or absence of wind, rainfall level) and the time elapsed between defecation and sampling in order to determine whether fluctuations in environmental conditions and field constraints would have had a significant effect on the concentrations of FGM. To avoid contamination and deterioration, samples were not taken on rainy days, when defecation took place in water or when individuals had diarrhoea. Indeed, after defecation, several factors, such as temperature, humidity, and other environmental conditions, may influence concentrations of FGM in the sample [57,58,59]. Finally, only the faeces sufficiently isolated from each other and therefore reliably identified were collected. A total of 199 faecal samples were thus obtained and 103 were selected for analyses. The choice of samples to analyse was made in such a way as to have an equivalent quantity of samples per individual (*n* = 7 ± 1), while making sure to have collection dates for each animal covering the entire period.

In collaboration with Professor Rupert Palme, the analysis of the concentration of FGM was carried out by the Physiology, Physiopathology and Experimental Endocrinology Unit of the University of Vienna (Austria). The protocol described in [60] and [57] was used for the extraction and measurement of FGM using the enzyme immune assays (EIA) technique. We obtained a value of FGM in ng/g per sample (Table 1). This protocol was checked and validated in many species including ungulates [61,62,63,64,65,66]. We performed the measures twice, on wet and dry faeces and the values were correlated at 98.18 % (r = 0.9818, *p* < 0.0001). We chose to use FGM values on wet faeces as there are fewer protocol steps for measurement. The detailed protocol is given in the Appendix A.

### 2.5. Statistical Analyses

We used eigenvector centrality, leadership score and mean FGM concentrations per week during the observation period. Age and dominance rank were stable. Therefore, we could not implement individual identities as random factor in the next analyses as it was collinear with age and dominance rank (VIF > 4). Indeed, the potential collinearity between our predictor variables was tested by the calculation of variance inflation factor (VIF) from the R package *car*. In our different models (except Model 2, see below), age and dominance rank always showed a VIF > 4, meaning that they were highly correlated (adj r^2^ = 0.79, F = 388, *p* < 0.0001, Figure 1a). Therefore, we decided to remove dominance rank as variable in the models 1, 3 and 4 as it is dependent on age, which is also shown in previous studies [30,31,38,67] but we tested it in a separate model. Subsequently, multifactorial linear models (LM) were applied as follows:FGM—Age × Leadership + Age × Eigenvector + Leadership × Eigenvector (Model 1)Rank—Age × FGM (Model 2)Eigenvector—Age × FGM (Model 3)Leadership—Age × FGM + Eigenvector × FGM (Model 4)Dominance—Age × FGM (Model 5)

We tested leadership as influenced by the eigenvector centrality as usually done in studies on leadership [68,69,70], by using the function *lm()* from the R package *car* [71]. Age was implemented as continuous variable. *p*-values for LM were calculated by *Monte Carlo sampling* with 10,000 permutations, using the function *PermTest()* of the R package *pgirmess* [72]. Permutation tests for LM are well adjusted for moderate sample size and do not require normal distribution of model residuals [73]. We repeated three times the tests to check for the stability of the statistics (α = 0.05). All statistical analyses were performed with R, version 4.0.3 (R Core Team) [74]. Data as detailed analyses are available on Zenodo repository: https://doi.org/10.5281/zenodo.6348624 (accessed on 14 February 2022).

## 3. Results

Results of Model 1 (adj r^2^ = 0.79, F = 67.29, *p* < 0.0001) show that the FGM is influenced negatively by age (*p* < 0.0001, Figure 1b,c), negatively by eigenvector centrality (*p* = 0.002, Figure 1b), and by the interaction Age × Eigenvector (*p* < 0.0001, Figure 1b) and the interaction Age × Leadership (*p* = 0.0044, Figure 1c). For the interaction Age × Eigenvector, in subadults, the effect is inverse compared to adults and juveniles, meaning that in subadults, FGM concentrations increase with eigenvector centralities. For the interaction Age × leadership, there was no initiations in juveniles and subadults, so the leadership effect was absent contrary to adults. Leadership and Leadership × Eigenvector did not influence the FGM (respectively *p* = 0.341 and *p* = 0.172).

Results of Model 2 (adj r^2^ = 0.81, F = 150.2, *p* < 0.0001) indicate that the dominance rank is positively influenced by age of individuals (the higher the age, the higher the rank, *p* < 0.0001, Figure 1a), negatively by FGM (*p* = 0.006, Figure 1d) with higher rank individuals having lower FGM concentrations and interaction Age × FGM (*p* = 0.026).

Results of Model 3 (adj r^2^ = 0.18, F = 8.75, *p* < 0.0001) show that the eigenvector centrality is negatively influenced by FGM (*p* = 0.006), interaction Age × FGM (*p* < 0.0001) but not by age alone (*p* = 0.408).

Results of Model 4 (adj r^2^ = 0.69, F = 45.94, *p* < 0.0001) indicate that the leadership score is influenced positively by age (*p* < 0.0001) but not by other factors or interactions (*p* > 0.353).

Finally results of Model 5 (adj r^2^ = 0.81, F = 150.2, *p* < 0.0001) indicate that dominance is still linked strongly and negatively by FGM concentration (*p* = 0.006) even if age is included in the model (*p* < 0.0001). The interaction Age × FGM also influenced dominance.

## 4. Discussion

In this study, we found that the FGM concentration [75,76], is linked to different variables of social status of European bison, specifically age, dominance rank, eigenvector centrality but also to several variable interactions.

We had to remove dominance rank from three of our models, because this factor is highly determined by age in European bison [38] but also in other bovines [30,31,67]. However, as shown in many studies, dominance rank is linked to glucocorticoid concentrations in one way [5,6,7], or the other [8,9]. Two models are opposed concerning the sense of the link between rank and glucocorticoids concentrations. If subordinate animals are subjected to aggression leading to chronic glucocorticoids concentration elevation, the ‘stress of subordination’ hypothesis predicts that the concentrations will be higher in subordinates than dominants (e.g., stress of dominance). If dominant individuals are subject to pressures to maintain their position and to fight, high dominant individuals will show a high concentration of glucocorticoids concentration (e.g., stress of domination) [7]. For instance, in captive animals, there is more aggression from the dominant towards the subordinate for whom the inability to escape results in an increase of the glucocorticoid hormone secretions [4,5]. Under natural conditions, a single study, carried out on olive baboons (*Papio anubis*), showed that the basal concentration of glucocorticoids was also higher in subordinate individuals [77], while five other studies, grouping together several taxa, demonstrated a higher concentration of glucocorticoids concentration in dominant individuals [78,79,80,81,82]. Thus, according to the literature, even if there is no absolute consensus on the question [11], a high secretion of glucocorticoids is more generally observed in the dominant than in the subordinate individuals, in particular for species living in stable social groups [6]. According to some authors, the high concentration of glucocorticoids in the dominant individual is not directly due to hierarchical status, but to agonistic behaviours expressed by the individual that allow him to access and maintain such a position [83,84]. In our study, we found that subordinate individuals have higher FGM concentrations than the dominant ones, but this is strongly mediated by age as subordinate individuals are younger individuals [38]. It is thus difficult to conclude about whether age or dominance is the prior factor affecting glucocorticoids, whilst studies have shown that, in same age groups of individuals, a correlation between dominance and FGM concentrations still existed [85,86]. However, given the results we got on dominance interactions with many unidirectional agonistic behaviours from dominant to subordinate, we may suggest that the stress of subordination exists in bison. Such group composition experiments would be interesting to test on bison, but this species being wild and vulnerable, is not manageable like domestic groups. Even if animals are transferred and group composition change in captive bison, researchers cannot decide about the group composition as they do with cattle.

A similar influence is found on the FGM concentration with the interaction between leadership and age showing that individuals who initiate more successfully show lower glucocorticoids concentration compared to others. This result can be explained by the greater experience and knowledge of older individuals who cope better with the uncertainty of changing location and resource availability [13]. The other explanation could be that individuals initiating movements do it because they feel the need to do so [35,42,68] and by initiating these movements, they thus fulfil their needs and show lower glucocorticoids concentration. One may think that leaders are more stressed because of their position [87,88], but in humans, a study showed that leaders have lower concentrations of glucocorticoids probably due to a sense of control [89].

Finally, we found that the preferential leaders in bison, i.e., the older and more dominant individuals which are also the more central ones [38,42], have lower FGM concentrations compared to other group members. Other factors with a potential influence such as reproductive status, the transmission of social rank by previous generations, season (and consequently resource availability) or even parasitic infections should be studied further as these factors are known to have an effect on social centrality, dominance rank and even leadership. How FGM concentration is a mediator between these factors is important to assess.

In addition, we found a link between the FGM concentrations and the eigenvector centrality. More central individuals have lower FGM concentration, except in subadults but this concerns only two individuals. This social centrality is based on spatial proximity and thus is more a spatial centrality. Animals being more spatially central in the group can feel more protected [90,91]. This correlation is also found with species showing grooming behaviour as central individuals are more groomed and this grooming decreases glucocorticoids concentration [17]. Socially isolated individuals also show higher glucocorticoids concentration [20] as well in cattle [92,93] as in other species [94,95] including humans [96]. The relationship between FGM and eigenvector centrality seems to be mediated by the age of individuals. In fact, Ramos et al. [38,42] showed that juveniles have similar centralities than adults because of their dependency but subadults have lower ones. The interaction here shows that FGM concentration decreases with centrality in juveniles and adults but not in subadults. The subadults often have higher glucocorticoid concentrations because they are less protected by their mother, especially if the mother has a new calf, need to find a place in the group and are subordinates.

Of course, this study has some limitations, the most important is having about seven samples per individual in eleven weeks of study. This sample size should be increased, but results show consistent concentrations of FGM per individual (see for instance Figure 1d). Importantly, future studies should focus on stressful events that could affect the link between FGM concentrations and social status that we found in our study. This could help in better understanding the mechanisms underlying social life of this species.

## 5. Conclusions

All these results are not only interesting for a better understanding of links between physiological and behavioural mechanisms in animal societies, but also for European bison management itself. Indeed, the species is listed as a threatened species on the IUCN Red List and requires especially high attention in conservation and rewilding proceedings [13,97,98,99]. Measurement of faecal glucocorticoid metabolites, associated with other parameters and particularly in response to stressors, could thus be a valuable tool to follow and improve the adaptation and the welfare of herds in the semi-wild and in captivity [39]. This new knowledge could, for instance, allow us to take into account the importance of older females as group members as being stable and reassuring figures during captures, veterinary interventions or new reintroductions [13,38]. Indeed, it has been shown in elephants or primates that these individuals attract attention of other individuals and are recognised as knowledge repository [33,100]. This could have a direct positive impact on survival of the herd as shown in elephants [101].

## Figures and Tables

**Figure 1 animals-12-00849-f001:**
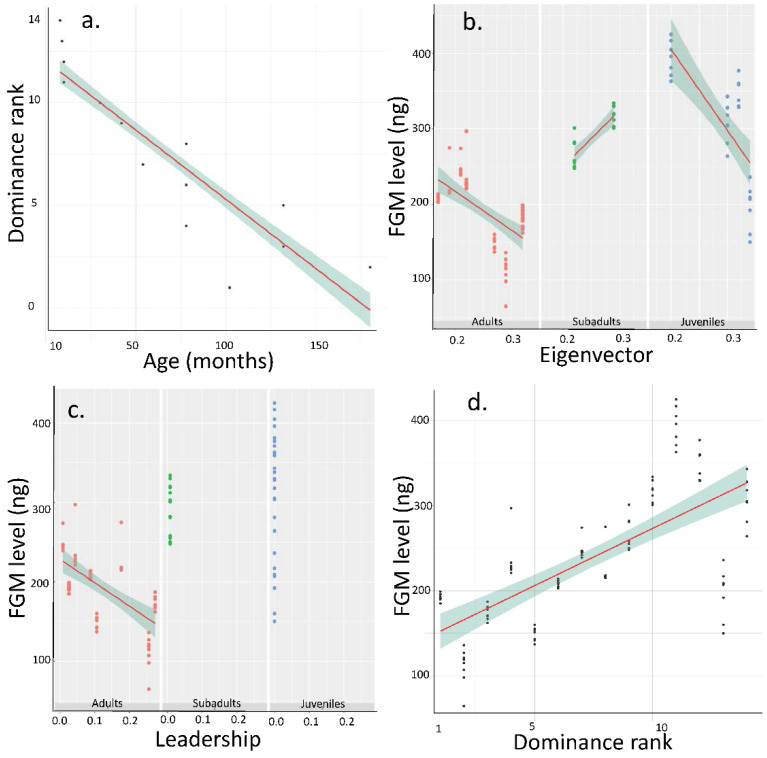
Main effects obtained between faecal glucocorticoid metabolites (FGM, ng/g-1) and social status in European bison. (**a**) Dominance in function of Age, (**b**) FGM concentration in function of Eigenvector (arbitrary unit) and Age (months), (**c**) FGM concentration in function of leadership (relative frequency) score and Age, (**d**) FGM concentration in function of dominance rank. Higher rank individual is ranked 1 whilst lower rank individual is ranked 14.

**Table 1 animals-12-00849-t001:** Individual measures for the total observation period. Higher rank individual is ranked 1 whilst lower rank individual is ranked 14. A.U. means arbitrary unit.

Individual	Age (Months; Years)	Eigenvector (A.U.)	Dominance Rank	Leadership (Relative Ratio)	FGM Mean ± SD (ng/g)
Kareta (Kr)	180; 15	0.29	2	0.23880597	272 ± 89
Kaga (Kg)	132; 11	0.27	5	0.097014925	297 ± 78
Katarina (Kt)	132; 11	0.32	3	0.253731343	210 ± 52
Katona (Ko)	102; 8.5	0.32	1	0.02238806	187 ± 47
Ina (In)	78; 6.5	0.17	6	0.104477612	177 ± 70
Moesja (Mo)	78; 6.5	0.19	8	0.171641791	264 ± 73
Wisha (Wi)	78; 6.5	0.22	4	0.052238806	227 ± 47
Frida (Fr)	54; 4.5	0.21	7	0.014925373	200 ± 44
Kristy (Ky)	42; 3.5	0.22	9	0.029850746	216 ± 93
Neréna (Nr)	30; 2.5	0.29	10	0.014925373	216 ± 49
Nelson (Nl)	10; 0.83	0.32	12	0	309 ± 80
Némar (Nm)	10; 0.83	0.2	11	0	283 ± 101
Nejen (Nj)	9; 0.72	0.34	13	0	275 ± 74
Névita (Nv)	8; 0.67	0.3	14	0	199 ± 42

## Data Availability

Data are available on Zenodo: https://doi.org/10.5281/zenodo.6348624 (accessed on 14 February 2022).

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
