# Peer review of "Glucocorticoids of European Bison in Relation to Their Status: Age, Dominance, Social Centrality and Leadership"

_animals, 2022, doi:10.3390/ani12070849_

Round 1

Reviewer 1 Report

This manuscript attempts to evaluate the association between social status and faecal glucocorticoid metabolites. The authors use a combination of behavioural observations, social network analysis and faecal hormone assays to do this. The authors conclude that older, more dominant females are less ‘stressed’, and that they are important leaders in European bison herds.

I believe that there are a number of issues with how the authors discuss stress, and the role that glucocorticoids play in the stress response. I would argue that a significant rewrite of the introduction is required, and a revision of the discussion too. Moreover, I believe that the authors could include additional factors into their models, to more adequately investigate social stress.  

The primary issue with the introduction as it is written as if glucocorticoids and stress are interchangeable. This is not the case, and the following references describe this issue well:

  • MacDougall-Shackleton, S. A., Bonier, F., Romero, L. M., & Moore, I. T. (2019). Glucocorticoids and “stress” are not synonymous. Integrative Organismal Biology, 1(1), obz017.
  • Palme, Rupert. "Non-invasive measurement of glucocorticoids: Advances and problems." Physiology & Behavior 199 (2019): 229-243.
  • Breuner, C. W., Delehanty, B., & Boonstra, R. (2013). Evaluating stress in natural populations of vertebrates: total CORT is not good enough. Functional Ecology, 27(1), 24-36.

Moreover, while an acute increase in glucocorticoids in response to a stressor might indicate that an individual is dealing with more stressors, long-term glucocorticoid profiles are not simply higher in more ‘stressed’ animals, and lower in less ‘stressed’ animals. (see Dickens, M. J., & Romero, L. M. (2013). A consensus endocrine profile for chronically stressed wild animals does not exist. General and comparative endocrinology, 191, 177-189 for a review).

The authors should expand on their introduction on what the stress response is, the role that glucocorticoids play in it, as well as expanding on the role that social hierarchy and structure can play on glucocorticoids/the individuals who are exposed to more social stressors. Indeed, the authors skip over how social structure can affect glucocorticoids in dominant/subordinate individuals, just saying that in some publications one trend is seen, while in others, the opposite is seen. A more thorough review of the literature is required. With this in mind, it would be helpful to the reader to understand more about how social dominance is achieved and maintained in bison. Is a high rank simply achieved by age, or must females physically/psychologically intimidate subordinates? This will also inform their hypothesis.  

Furthermore, while the authors include dominance rank in their models, without information on dominance interactions, it’s difficult to understand how individuals might be more/less ‘stressed’. For example, more dominant individuals might have lower glucocorticoids simply because they are challenged less frequently than subadults. For this reason, I believe it would be helpful to include the number of agonistic encounters an individual is involved in, regardless of loss/win as a variable in their models. It is reasonable to think that individuals who are more involved in conflict behaviour would generally have higher glucocorticoid concentrations. This might serve as a proxy for how socially challenged an individual might be/how unstable their position is.

Line by line comments follow. I have refrained from adding too many comments to the introductory paragraphs as it is my opinion that it needs to be rewritten:

  • L12 – “.. to different aspects of social…”
  • L14 – “associated with other parameters” – this s vague and an unnecessary addition at this stage.
  • L15 – you mention that monitoring glucocorticoids could improve adaptation of bison here and further on in the manuscript. How so?
  • L15-18 – this is neither a summary of your results, nor do I believe that your results tell you that dominant females are reassuring to the rest of the group nor that they are more able to respond to stressors. Dominant females might be better able to cope with veterinary procedures/translocations etc, but as you don’t measure the glucocorticoid response to a stressor, you can’t tell this. Moreover, you do not test how subordinates respond behaviourally/physiologically to stressors in the presence of, and without dominant individuals so I don’t believe you can make a statement on reassuring others.
  • L19+33– “better survive”. Stress does not automatically link to fitness. Too much stress could cause immunosuppression/death etc. This should be removed or revised.
  • L20: “..dominant individual in the group..”
  • L24: GCs are not a proxy for stress.
  • L26: “.. to interactions between the variables..”
  • L26: I am unsure what you mean by glucocorticoids mediating social life. How so?
  • L27: delete “also the”
  • L28-29: The concluding sentence seems to be on a tangent. It’s the first you talk about conservation in the abstract, and the paper doesn’t really expand on how this research improves species conservation. I don’t doubt that it could, but it’s not clear to the reader at any point.
  • L44-47: the paper on Prezwalski’s horses does not discuss a cause and effect between increased glucocorticoids and the stallion taking a position at the top. “However, on the enlargement day, the alpha male showed the highest increase of excreted GCMs. A reason for that could have been the change of his spatial position in the group. He was in the front position during most group movements and therefore behaved in an unusual way, as the alpha stallion normally walks behind his group”.
  • L51: it might be helpful to include a paragraph reviewing the interaction between social centrality (or even just SNA) and glucocorticoids, prior to discussing social centrality as you do for leadership and dominance. For example:
    • Brent, L. J. N., Semple, S., Dubuc, C., Heistermann, M., & MacLarnon, A. (2011). Social capital and physiological stress levels in free-ranging adult female rhesus macaques. Physiology & Behavior, 102(1), 76-83.
    • Wittig, R. M., Crockford, C., Lehmann, J., Whitten, P. L., Seyfarth, R. M., & Cheney, D. L. (2008). Focused grooming networks and stress alleviation in wild female baboons. Hormones and behavior, 54(1), 170-177.
    • Crockford, C., Wittig, R. M., Whitten, P. L., Seyfarth, R. M., & Cheney, D. L. (2008). Social stressors and coping mechanisms in wild female baboons (Papio hamadryas ursinus). Hormones and behavior, 53(1), 254-265.
  • L57: You’re not measuring the ‘stress level’.
  • L64-65: expand on why leadership and sociality in this species makes you think this. The reader so far knows very little about bison.
  • L80: delete “Further on”
  • L84: “.. at an individual…”
  • L86: is it normal not to have any adult males in the group? The social groups in reference 21 all have adult males in them. Might that attribute to differences in the results of your and their study, in particular with regards to age/dominance rank? Expand on this in your discussion.
  • L89: “..during any time of year.”
  • L96-100: Rephrase to “We observed the bison at a minimum distance of 50 metres. To avoid stressing the animals and to ensure that our presence did not interfere with natural activity patterns, this distance sometimes decreased slightly due to terrain geography or herd movement direction.”
  • L106: “Every 5 minutes.”
  • L110: define eigenvector centrality and how it is calculated.
  • L114: delete “for details).”
  • L118: should the de Vries reference just be a number?
  • L122: This section should be called ‘Leadership’.
  • L134: “..faecal samples..”
  • L135: the time after defecation that the samples were collected should be moved here.
  • L136: “the filled tubes were labelled with the individual ID and date and then…”
  • L147: you’ve not included body condition in your study so I don’t think this is relevant.
  • L154: this is a really small sample size considering your study took place over 11 weeks. How might this affect your results? You might have missed a stressful event for the herd that could have affected your results. Expand on this in the discussion.
  • L156: should Prof Palme, or a representative from the lab be a co-author if they produced results?
  • L159: EIA should first be spelled out prior to having the abbreviation.
  • L159: which antibody did you use? Was it for a specific metabolite? You mention cortisol in your abstract and conclusion but not here, nor in your supplementary information. More information about the EIAs should be included here.
  • L160: should be ng/g only.
  • L161: replace “approved covering” with “validated in”. Has this assay previously been validated for this species?
  • L164: fewer, not less.
  • L167: “mean FGM concentrations…”
  • L168: what did you use these variables for?
  • L168: did you enter age as a continuous or categorical factor in your models?
  • L177-180: please expand on why you chose the models that you did. Also include the full results in your results section.
  • L180: is there an association between leadership and rank?
  • L181-182: why did you do the opposite of other studies?
  • Results in general: please describe your results. Are associations positive? Negative?
  • Table 1: I’m not sure how necessary this table is? I would replace it with the results from your model.
    • What does A.U. stand for?
    • Units for FGM are not correct. Also, is the ± SE? SD?
  • Figure 1: ‘FGM level’ should be “FGM concentration (ng/g)”
  • L221: subordinate, not dominated.
  • L238-239: you could compare FGM concentrations across groups with different social structures though.
  • L241: but not in subadults…
  • L247+254: write out the formula in your discussion i.e. “the interaction between FGM and eigenvector centrality..”.
  • L266: you don’t show this – only that they have lower GCs. They might be better able to cope with stressors due to experience or having the support of their mothers, or they might encounter fewer stressors.
  • L266-269: why? Explain these statements.
  • L273: listed as what?
  • L279: do you show that these leaders are reassuring to others?
  • References: please check through your references as some are incomplete e.g. ref 4, ref 19, and journal names are inconsistent – some are abbreviated, some have full n.

Author Response

Reviewers 1 response to comments.

Q: This manuscript attempts to evaluate the association between social status and faecal glucocorticoid metabolites. The authors use a combination of behavioural observations, social network analysis and faecal hormone assays to do this. The authors conclude that older, more dominant females are less ‘stressed’, and that they are important leaders in European bison herds.

I believe that there are a number of issues with how the authors discuss stress, and the role that glucocorticoids play in the stress response. I would argue that a significant rewrite of the introduction is required, and a revision of the discussion too. Moreover, I believe that the authors could include additional factors into their models, to more adequately investigate social stress. 

A: Thank you very much for your comments. The manuscript is well enhanced thanks to them.

Q: The primary issue with the introduction as it is written as if glucocorticoids and stress are interchangeable. This is not the case, and the following references describe this issue well:

MacDougall-Shackleton, S. A., Bonier, F., Romero, L. M., & Moore, I. T. (2019). Glucocorticoids and “stress” are not synonymous. Integrative Organismal Biology, 1(1), obz017.

Palme, Rupert. "Non-invasive measurement of glucocorticoids: Advances and problems." Physiology & Behavior 199 (2019): 229-243.

Breuner, C. W., Delehanty, B., & Boonstra, R. (2013). Evaluating stress in natural populations of vertebrates: total CORT is not good enough. Functional Ecology, 27(1), 24-36.

Moreover, while an acute increase in glucocorticoids in response to a stressor might indicate that an individual is dealing with more stressors, long-term glucocorticoid profiles are not simply higher in more ‘stressed’ animals, and lower in less ‘stressed’ animals. (see Dickens, M. J., & Romero, L. M. (2013). A consensus endocrine profile for chronically stressed wild animals does not exist. General and comparative endocrinology, 191, 177-189 for a review).

A: We added the following paragraphs in the introduction “Many studies stipulated that stress and glucocorticoids level are interchangeable whilst this is not always the case [21–23]. Moreover, while an acute increase in glucocorticoids in response to a stressor might indicate that an individual is dealing with more stressors, long-term glucocorticoid profiles are not simply higher in more ‘stressed’ animals, and lower in less ‘stressed’ animals. This is why in the rest of the paper, we only mention faecal glucocorticoid metabolites (FGM) level. » and replaced all mentions of stress in the rest of the paper by glucocorticoid level.

Q: The authors should expand on their introduction on what the stress response is, the role that glucocorticoids play in it, as well as expanding on the role that social hierarchy and structure can play on glucocorticoids/the individuals who are exposed to more social stressors. Indeed, the authors skip over how social structure can affect glucocorticoids in dominant/subordinate individuals, just saying that in some publications one trend is seen, while in others, the opposite is seen. A more thorough review of the literature is required. With this in mind, it would be helpful to the reader to understand more about how social dominance is achieved and maintained in bison. Is a high rank simply achieved by age, or must females physically/psychologically intimidate subordinates? This will also inform their hypothesis. 

A: we added the following paragraphs: “Glucocorticoids levels are also linked to social capital but this correlation – positive or negative - depends on the stability of the hierarchy and other parameters as the reproduction state [17–20]. Mainly, high-ranking individuals have lower levels of glucocorticoids when the hierarchy is stable but higher levels when it is unstable. Having a certain number of relationships is good but too many may lead to negative effects on health. In Plains Bison (Bison bison, L. 1758), individuals have linear hierarchies and males tend to be more aggressive when growing older, using physical aggression. High-ranking males defend their position leading to stress of domination (e.g. Higher ranks are more stressed). However, females rely more on non-physical aggression, meaning intimidation ad this leads to stress of dominance (lower ranks are more stressed). As for other ungulates, rank is often linked to age, as to body weight, also linked to age.”

Furthermore, while the authors include dominance rank in their models, without information on dominance interactions, it’s difficult to understand how individuals might be more/less ‘stressed’. For example, more dominant individuals might have lower glucocorticoids simply because they are challenged less frequently than subadults. For this reason, I believe it would be helpful to include the number of agonistic encounters an individual is involved in, regardless of loss/win as a variable in their models. It is reasonable to think that individuals who are more involved in conflict behaviour would generally have higher glucocorticoid concentrations. This might serve as a proxy for how socially challenged an individual might be/how unstable their position is.

Indeed. We added the following sentences:

- “The linearity of the dominance is very high showing few bidirectional directions and few aggression from subordinate to dominant individuals. Never the subadult or juvenile bison show aggressive behaviours towards the adult ones. Up on 956 interactions, 141 were horn punches, 278 were threatening, 526 were supplanting behaviours and only eleven were avoiding behaviours. Detailed interactions are given in Zenodo repository: https://doi.org/10.5281/zenodo.6348624”

- “Two models are opposed concerning the sense of the link between rank and glucocorticoids levels. If subordinate animals are subjected to aggression leading to chronic glucocorticoids level elevation, the ‘stress of subordination’ hypothesis predicts that the levels will be higher in subordinates than dominants (e.g. stress of dominance). If dominant individuals are subject to pressures to maintain their position and to fight, high dominant individuals will show a high level of glucocorticoids level (e.g. stress of domination).”

- “However, given the results we got on dominance interactions with many unidirectional agonistic behaviours from dominant to subordinate, we may suggest that the stress of subordination exists in bison.”

Line by line comments follow. I have refrained from adding too many comments to the introductory paragraphs as it is my opinion that it needs to be rewritten:

Q: L12 – “.. to different aspects of social…”

A: Done

Q: L14 – “Associated with other parameters” – this s vague and an unnecessary addition at this stage.

A: removed

Q: L15 – you mention that monitoring glucocorticoids could improve adaptation of bison here and further on in the manuscript. How so?

A: We added “by long-term following states of animals and changing conditions according to them”

Q: L15-18 – this is neither a summary of your results, nor do I believe that your results tell you that dominant females are reassuring to the rest of the group nor that they are more able to respond to stressors. Dominant females might be better able to cope with veterinary procedures/translocations etc, but as you don’t measure the glucocorticoid response to a stressor, you can’t tell this. Moreover, you do not test how subordinates respond behaviourally/physiologically to stressors in the presence of, and without dominant individuals so I don’t believe you can make a statement on reassuring others.

A: We changed to “Dominant females might be better able to cope with veterinary procedures/translocation and might be used as models or helps for younger individuals in conservation programs.”

Q: L19+33– “better survive”. Stress does not automatically link to fitness. Too much stress could cause immunosuppression/death etc. This should be removed or revised.

A: Done

Q: L20: “..dominant individual in the group..”

A: Done

Q: L24: GCs are not a proxy for stress.

A: We removed this part in the abstract and discussed it in the introduction.

Q: L26: “.. to interactions between the variables..”

A: Done

Q: L26: I am unsure what you mean by glucocorticoids mediating social life. How so?

A: We removed the sentence

Q: L27: delete “also the”

A: Done

Q: L28-29: The concluding sentence seems to be on a tangent. It’s the first you talk about conservation in the abstract, and the paper doesn’t really expand on how this research improves species conservation. I don’t doubt that it could, but it’s not clear to the reader at any point.

A: We added “by collecting data on FGM and other social variables and adapt group composition or environmental conditions (e.g. supplement in food) according to the FGM levels of herd individuals.”

Q: L44-47: the paper on Przewalski’s horses does not discuss a cause and effect between increased glucocorticoids and the stallion taking a position at the top. “However, on the enlargement day, the alpha male showed the highest increase of excreted GCMs. A reason for that could have been the change of his spatial position in the group. He was in the front position during most group movements and therefore behaved in an unusual way, as the alpha stallion normally walks behind his group”.

A: we changed the sentence to: “However, a study in Przewalski's horse species, Equus ferus przewalskii, showed a link between the change in the structure of the enclosure, the secretion of glucocorticoids in the male and the lead of collective movements [15].

L51: it might be helpful to include a paragraph reviewing the interaction between social centrality (or even just SNA) and glucocorticoids, prior to discussing social centrality as you do for leadership and dominance. For example:

Brent, L. J. N., Semple, S., Dubuc, C., Heistermann, M., & MacLarnon, A. (2011). Social capital and physiological stress levels in free-ranging adult female rhesus macaques. Physiology & Behavior, 102(1), 76-83.

Wittig, R. M., Crockford, C., Lehmann, J., Whitten, P. L., Seyfarth, R. M., & Cheney, D. L. (2008). Focused grooming networks and stress alleviation in wild female baboons. Hormones and behaviour, 54(1), 170-177.

Crockford, C., Wittig, R. M., Whitten, P. L., Seyfarth, R. M., & Cheney, D. L. (2008). Social stressors and coping mechanisms in wild female baboons (Papio hamadryas ursinus). Hormones and behaviour, 53(1), 254-265.

A: We added these sentences: “Glucocorticoids levels are also linked to social capital but this correlation – positive or negative - depends on the stability of the hierarchy and other parameters as the reproduction state [16–19]. Having a certain number of relationships is good but too many may lead to negative effects on health.”

Q: L57: You’re not measuring the ‘stress level’.

A: We added a paragraph to discuss this point and change to FGM all along the manuscript after this paragraph.

Q: L64-65: expand on why leadership and sociality in this species make you think this. The reader so far knows very little about bison.

A: We added the sentence: “Indeed, in stable hierarchy or stable group composition, individuals with higher dominance rank and higher social centrality often show lower FGM level [16–19]. Dominance is also linked to spatial position of group members and low or middle rank individuals are often at the group spatial periphery, which may increase their FGM level [24]. Moreover, leadership is associated with a more independent behaviour which is linked to lower levels of FGM [37–40].”

Q: L80: delete “Further on”

A: Done

Q: L84: “.. at an individual…”

A: Done

Q: L86: is it normal not to have any adult males in the group? The social groups in reference 21 all have adult males in them. Might that attribute to differences in the results of your and their study, in particular with regards to age/dominance rank? Expand on this in your discussion.

A: We added the sentences: “As for many ungulates, bison show sexual segregation with males only coming in the herd during the reproduction season [41–43]. Even in semi-wild herds having a male, this one is often isolated.”

Q: L89: “..during any time of year.”

A: Done

Q: L96-100: Rephrase to “We observed the bison at a minimum distance of 50 metres. To avoid stressing the animals and to ensure that our presence did not interfere with natural activity patterns, this distance sometimes decreased slightly due to terrain geography or herd movement direction.”

A: Done

Q: L106: “Every 5 minutes.”

A: Done

Q: L110: define eigenvector centrality and how it is calculated.

A: We added: “This measures the number and strength of associations of an individual, whilst taking into account the strength of associations of the individuals with whom it is itself associated and is often described as popularity [46].”

Q: L114: delete “for details).”

A: Done

Q: L118: should the de Vries reference just be a number?

A: Corrected

Q: L122: This section should be called ‘Leadership’.

A: Done

Q: L134: “..faecal samples..”

A: Done

Q: L135: the time after defecation that the samples were collected should be moved here.

A: Done

Q: L136: “the filled tubes were labelled with the individual ID and date and then…”

A: Done

Q: L147: you’ve not included body condition in your study so I don’t think this is relevant.

A: removed

Q: L154: this is a really small sample size considering your study took place over 11 weeks. How might this affect your results? You might have missed a stressful event for the herd that could have affected your results. Expand on this in the discussion.

A: Indeed, we added a paragraph in the discussion: “Of course, this study has some limitations, this most important being having about seven samples per individual on eleven weeks of study. This sample size should be increased, but results show consistent levels of FGM per individual (see for instance fig.1d). Importantly, future studies should focus on stressful events that could affect the link between FGM levels and social status we found in our study. This could conduct to better understand the mechanisms underlying social life of this species.”

Q: L156: should Prof Palme, or a representative from the lab be a co-author if they produced results?

A: No, it was an agreement with Prof Palme since the beginning of the study.

Q: L159: EIA should first be spelled out prior to having the abbreviation.

A: Done

Q: L159: which antibody did you use? Was it for a specific metabolite? You mention cortisol in your abstract and conclusion but not here, nor in your supplementary information. More information about the EIAs should be included here.

A: the detailed protocol is given in Supplementary material, didn’t you get access to it?

Q: L160: should be ng/g only.

A: Sorry, done

Q: L161: replace “approved covering” with “validated in”. Has this assay previously been validated for this species?

A: Done. FGM was measured twice in American bison but this is the only study in bison bonasus.

Q: L164: fewer, not less.

A: Done

Q: L167: “mean FGM concentrations…”

A: Done

Q: L168: what did you use these variables for?

A: Sorry, but we don’t understand the question. Could you please be more precise?

Q: L168: did you enter age as a continuous or categorical factor in your models?

A: continuous, we precised it.

Q: L177-180: please expand on why you chose the models that you did. Also include the full results in your results section.

A: Data as detailed analyses are available on Zenodo repository: 10.5281/zenodo.6330534

Q: L180: is there an association between leadership and rank?

A: Yes, but this is irrelevant as rank is dependent on age.

Q: L181-182: why did you do the opposite of other studies?

A: sorry, this is a syntax mistake, we changed the sentence.

Q: Results in general: please describe your results. Are associations positive? Negative?

A: Done

Q: Table 1: I’m not sure how necessary this table is? I would replace it with the results from your model.

A: What do you mean by the results of the model? The models given in Zenodo are fully described here. As we used permutations, there is only a P-value given.

Q: What does A.U. stand for?

A: Arbitrary Unit. We precised it.

Q: Units for FGM are not correct. Also, is the ± SE? SD?

A: SD, we added it.

Q: Figure 1: ‘FGM level’ should be “FGM concentration (ng/g)”

A: done

Q: L221: subordinate, not dominated.

A: Done

Q: L238-239: you could compare FGM concentrations across groups with different social structures, though.

A: Yes, in theory, but contrary to highland cattle or cows where groups can be easily managed to have for instance only females of same age, bison herds are only created at the moment for conservation as specified: “Such group composition experiments would be interesting to test on bison, but this species being wild and vulnerable, it’s not manageable like domestic groups.”

Q: L241: but not in subadults…

A: indeed, we added: “, excepted in subadults but this concerns only two individuals.”

Q: L247+254: write out the formula in your discussion i.e. “the interaction between FGM and eigenvector centrality..”.

A: Done

Q: L266: you don’t show this – only that they have lower GCs. They might be better able to cope with stressors due to experience or having the support of their mothers, or they might encounter fewer stressors.

A: corrected

Q: L266-269: why? Explain these statements.

A: we added “as these factors are known to have an effect on social centrality, dominance rank and even leadership. How FGM level is a mediator between these factors is important to assess.”

Q: L273: listed as what?

A: as threatened species. We precised it.

Q: L279: do you show that these leaders are reassuring to others?

A: We added the sentences: “Indeed, it has been shown in elephants or primates that these individuals attract attention of other individuals and are recognised as knowledge repository [26,94]. This could have a direct positive impact on survival of the herd as shown in elephants [95].”

Q: References: please check through your references as some are incomplete e.g. ref 4, ref 19, and journal names are inconsistent – some are abbreviated, some have full n.

A: checked. We are sorry, we used Zotero and we don’t know why there were some many mistakes.

Reviewer 2 Report

Dear Athors, I have some suggestions to make a few improvements of the paper. I will describe them by lines:

12 - age does not match to social status. It is more individual characteristic as you wrote in 49 line.

23 - this is the first time when you mentioned Latin name of the species, so add L. 1758.

30 - you should add 'glucocorticoids' to key words. it is important to correct scientific searching 

33 - [1,2] instead of [1,2] The same for whole text. 

40 - I would rather write 'basic' instead of 'basal'

51 - '...' no need

53 - maybe while instead of whilst

58-59 - like remarks to 23 line - add L. 1758, here Latin name is written first time in main text

75 - add L. 1758 to the mentioned species

77 -  the Yellowstone National Park (concrete),  whereas a national park instead of 'the National Parks'. Or put here a concrete name of this national park. English mistake.

79 - from September instead of September

82 - Study subjects

86 - 8 and 9 months

91 - twice et al. (Italics) instead of et al. 

92 - Data collection

96 - was instead of is

97 - whereas maximum distance was.... add this information

114, 118, 132 -  et al.

134 -  'The faeces samples were collected during the observations'. It is not written but if I understand this correctly, you collect faeces from concrete observed individuals right?? 

166 - analysis

203 - Table 1 - In truth I don't like age of animals written in months... more visible and clear is for example, 2,5 years old. On a graph 1a months are correct, but not here in the table. 

All data in the table should be centered

207 - age. Maybe put a. ..., b. .... etc. instead of a. , b.  It will be more visible.

218-220 - this statements is a perfect repetition of statement from 35-37 lines. Decide which part will be better for it and delete one. 

248 - et al. 

285 - CS?

312 - why you put short name of Anim Beh, whereas in 307 line Functional Ecology instead of Funct Ecol? It should be the same style for each journal name. Use short names everywhere in whole references.

312, 314 - what is more - one time Anim Beh, one time Animal Behaviour. Why?

319 -  when you write Canadian Journal of zoology , each word of the journal name should be written in a big letter. Remember this is a proper way. Instead of this, put a short name. 405 and 406 line - the same mistake.

353 -  try to thicken the text in line above

357 - do we need this one: " (<italic>Bison Bonasus</Italic>)." ?

370  - why ETHOLOGY instead of Ethology? The same in 373 line.

Author Response

Reviewer 2 response to comments.

Q: Dear Authors, I have some suggestions to make a few improvements of the paper. I will describe them by lines:

A: Thank you very much for your comments.

Q: 12 - age does not match to social status. It is more individual characteristic as you wrote in 49 line.

A: Indeed, age is an individual characteristic but a cue for experience which is kind of social status as for elephants. We specified it.

Q: 23 - this is the first time when you mentioned Latin name of the species, so add L. 1758.

A: Done

Q: 30 - you should add 'glucocorticoids' to key words. it is important to correct scientific searching

A: Done

Q: 33 - [1,2] instead of [1,2] The same for whole text.

A: Done

Q: 40 - I would rather write 'basic' instead of 'basal'

A: Done

Q: 51 - '...' no need

A: Done

Q: 53 - maybe while instead of whilst

A: Done

Q: 58-59 - like remarks to 23 line - add L. 1758, here Latin name is written first time in main text

A: Done

Q: 75 - add L. 1758 to the mentioned species

A: Done

Q: 77 -  the Yellowstone National Park (concrete),  whereas a national park instead of 'the National Parks'. Or put here a concrete name of this national park. English mistake.

A: corrected

Q: 79 - from September instead of September

A: Done

Q: 82 - Study subjects

A: Done

Q: 86 - 8 and 9 months

A: Done

Q: 91 - twice et al. (Italics) instead of et al.

A: Done

Q: 92 - Data collection

A: Done

Q: 96 - was instead of is

A: Done

Q: 97 - whereas maximum distance was.... add this information

A: We changed to: “maximum distance of 60 metres but minimum distance of 50 metres”

Q: 114, 118, 132 -  et al.

A: Done

Q: 134 -  'The faeces samples were collected during the observations'. It is not written but if I understand this correctly, you collect faeces from concrete observed individuals right??

A: Right, we added this information

Q: 166 - analysis

A: done

Q: 203 - Table 1 - In truth I don't like age of animals written in months... more visible and clear is for example, 2,5 years old. On a graph 1a months are correct, but not here in the table.

A: corrected, we added age in years.

Q: All data in the table should be centered

A: Done

Q: 207 - age. Maybe put a. ..., b. .... etc. instead of a. , b.  It will be more visible.

A: Done

Q: 218-220 - this statements is a perfect repetition of statement from 35-37 lines. Decide which part will be better for it and delete one.

A: Done

Q: 248 - et al.

A: Done

Q: 285 - CS?

A: Done

Q: 312 - why you put short name of Anim Beh, whereas in 307 line Functional Ecology instead of Funct Ecol? It should be the same style for each journal name. Use short names everywhere in whole references. 312, 314 - what is more - one time Anim Beh, one time Animal Behaviour. Why? 319 -  when you write Canadian Journal of zoology , each word of the journal name should be written in a big letter. Remember this is a proper way. Instead of this, put a short name. 405 and 406 line - the same mistake. 353 -  try to thicken the text in line above 357 - do we need this one: " (<italic>Bison Bonasus</Italic>)." ? 370  - why ETHOLOGY instead of Ethology? The same in 373 line.

A: Sorry for these mistakes, we used Zotero, so the references list is automatically done and we don’t understand why some journals were not written correctly. We corrected that.

Round 2

Reviewer 1 Report

The manuscript is much improved and now reflects what the authors measured. I appreciate the time taken to respond to all of my comments. There are a few instances where comments have not been fully resolved.

Throughout: GCs and FGM should be a concentration, not a level

L12: age is a proxy for experience

L16-18: this is something that you say further research could tell you, but not something that you were able to determine from your study. Therefore it’s not relevant as a summary of your findings.

L49-51: Explain why the link is relevant. Interventions and perceived stressors (illustrated by the GC increase) might change leadership properties in a social group.

L50: delete ‘I’ after enclosure.

L51: new paragraph starting ‘Glucocorticoids…’

L69-79: I would move this paragraph to L51 as it still discusses leadership.

L86-97: unless visitor presence and study site differences are relevant to ypur study, delete this sentence. If it is relevant, explain why.

L119: delete ‘during winter or any time of year’. If they are not supplemented, you don’t need to clarify which seasons they’re not supplemented during.

L137, 147, 161: why are your sampling methods in italics?

L158: thank you for adding a link to your data repository. The description of your study on Zenodo needs to be updated. It still refers to GCs as a proxy for stress and that GCs mediate life.

L261: This still doesn't make sense to me. GCs are associated with social variables, but you don't determine the relationship. Do more dominant individuals have lower GCs because they are better able to cope with stressors (low GCs leads to high rank), or because they are simply exposed to fewer stressors (high rank leads to low GCs)?

L292: clarify that this is not possible in wild bison, In captive bison you will have differences in the group composition, and animals are frequently transferred to new groups/released to the wild.

L294-308: the paragraphs on rank, leadership, and age should come before this as they’re so intrinsically linked. This would improve the flow.

L319-321: this is not contrary to your initial hypothesis “we made the following hypothesis: the oldest females in the herd, which are the preferential leaders and the more dominant and central individuals, should have a lower average rate of FGM compared to the other group members.“

L338: measuring GCs, in particularly in response to stressors could be valuable.

Supplementary materials: I still don’t see the specific antibody you used. You mention ‘antibody solution’. Did you use the 5β-androstane-3α-o1-11,17-dione-17-CMO:BSA antibody as stated in reference 32?

References: please check them again. Reference 37 for example still says ‘Undefined’

Author Response

Q: The manuscript is much improved and now reflects what the authors measured. I appreciate the time taken to respond to all of my comments. There are a few instances where comments have not been fully resolved.

A: Thank you very much for your positive comments. And sorry that we fully answered them previously. We tried to answer in this new version all our requests positively.

Q: Throughout: GCs and FGM should be a concentration, not a level

A: Done

Q: L12: age is a proxy for experience

A: Done

Q: L16-18: this is something that you say further research could tell you, but not something that you were able to determine from your study. Therefore it’s not relevant as a summary of your findings.

A: We removed this part from the simple summary.

Q: L49-51: Explain why the link is relevant. Interventions and perceived stressors (illustrated by the GC increase) might change leadership properties in a social group.

A: We added “indicating that environmental changes may influence leadership through glucocorticoids concentration. Thus, FGM might be a mediator to cope with external pressures.”

Q: L50: delete ‘I’ after enclosure.

A: Done

Q: L51: new paragraph starting ‘Glucocorticoids…’

A: Done

Q: L69-79: I would move this paragraph to L51 as it still discusses leadership.

A: Done

Q: L86-97: unless visitor presence and study site differences are relevant to ypur study, delete this sentence. If it is relevant, explain why.

A: We removed the sentence.

Q: L119: delete ‘during winter or any time of year’. If they are not supplemented, you don’t need to clarify which seasons they’re not supplemented during.

A: Done

Q: L137, 147, 161: why are your sampling methods in italics?

A: Because they are formal methods, but we removed the italics.

Q: L158: thank you for adding a link to your data repository. The description of your study on Zenodo needs to be updated. It still refers to GCs as a proxy for stress and that GCs mediate life.

A: Done.

Q: L261: This still doesn't make sense to me. GCs are associated with social variables, but you don't determine the relationship. Do more dominant individuals have lower GCs because they are better able to cope with stressors (low GCs leads to high rank), or because they are simply exposed to fewer stressors (high rank leads to low GCs)?

A: We removed that FGMs are a mediator.

Q: L292: clarify that this is not possible in wild bison, In captive bison you will have differences in the group composition, and animals are frequently transferred to new groups/released to the wild.

A: We added the sentence: “Even if animals are transferred and group composition change in captive bison, re-searchers cannot decide about the group composition as they do with cattle.” 

Q: L294-308: the paragraphs on rank, leadership, and age should come before this as they’re so intrinsically linked. This would improve the flow.

A: Done 

Q: L319-321: this is not contrary to your initial hypothesis “we made the following hypothesis: the oldest females in the herd, which are the preferential leaders and the more dominant and central individuals, should have a lower average rate of FGM compared to the other group members.“

 A: We removed “contrary to…”

Q: L338: measuring GCs, in particularly in response to stressors could be valuable.

 A: Thanks! We added “and particularly in response to stressors”.

Q: Supplementary materials: I still don’t see the specific antibody you used. You mention ‘antibody solution’. Did you use the 5β-androstane-3α-o1-11,17-dione-17-CMO:BSA antibody as stated in reference 32?

A: Yes it is, we added it.

Q: References: please check them again. Reference 37 for example still says ‘Undefined’

A: Done.
